



**Design methodology to determine water quality monitoring strategy of**
**surface water treatment plants**
Petra Ross*, Kim. van Schagen**,  Luuk. Rietveld*
* Delft University of Technology, PO Box 5048, 2600 GA Delft, the Netherlands, p.s.ross@tudelft.nl
** RoyalHaskoningDHV BV, PO Box 1132, 3800 BC Amersfoort, the Netherlands

9       **Abstract**
Primary goal of a drinking water company is to produce safe drinking water fulfilling the quality
standards defined by national and international guidelines. To ensure the produced drinking water meets
the quality standards, sampling of the drinking water is carried out on a regular (almost daily) basis. It
is the dilemma that the operator wishes to have a high probability of detecting a bias while minimizing
his measuring effort. In this paper a seven step design methodology is described on how to come to an
optimised water quality monitoring scheme. It was shown that the previous on-line monitoring program
of a WTP could be optimised. Besides using soft-sensors as surrogate sensors for parameters currently
not available on-line, they can also provide a cost effective alternative when used to determine multiple
parameters required through one single instrument.
**Keywords**
Data requirements; design methodology; model-based optimization; soft-sensors

## INTRODUCTION

Primary goal of a drinking water company is to produce safe drinking water fulfilling the quality standards defined
by national and international guidelines. To ensure the produced drinking water meets the quality standards,
sampling of the drinking water is carried out on a regular (almost daily) basis.

Common practice is that (drinking) water treatment plants (WTPs) are designed in such a robust way that the
effluent quality can be guaranteed without direct control on the incoming water quality (Vanrolleghem and Lee,
2003;Bosklopper et al., 2004). A WTP consists of several individual treatment steps placed in series, with every
treatment step being responsible for the removal (or addition) of certain compounds. All the interactions between
the processes ask for an integrated plant-wide approach, optimizing the effluent quality and operational costs
(Bosklopper et al., 2004;Nopens et al., 2010).

Van der Helm et al. (2008b) investigated three possible objectives for plant-wide optimization of operation of
existing WTPs and concluded that the objective for integrated optimization should be the improvement of water
quality and not a reduction in environmental impact and costs. The effects of these latter two are negligible
compared to the environmental impact and costs for the society as a whole when more bottled water is used for
drinking water as a result of insufficient (confidence in) tap water quality.

Direct control of water quality becomes more and more important as a result of more stringent criteria and the
deterioration of source water (Vanrolleghem and Lee, 2003;van Schagen et al., 2010). Especially WTPs that use
surface water as a source, experience increased pollution in the form of organic micropollutants and increased
organic matter concentrations present in the surface water bodies (Verliefde et al., 2007;Bertelkamp et al., 2014).
Besides, large fluctuations in water temperature and water quality can be noticed, which increases the need for
direct control of the WTP.

Nowadays, many WTPs are monitored and controlled by SCADA (Supervisory Control and Data Acquisition)
systems (Jansen et al., 1997). The functions of SCADA systems for WTPs include: (1) collection of on-line
measurement data, (2) surveillance of the measuring chain including operations and (3) process control and other
relevant operations (Gunatilaka and Dreher, 2003).  On-line measurements are the first indicators that give the



operators information about the state the plant is in. Besides on-line measurements, laboratory measurements are
taken at a regular interval to check that the produced drinking water meets the quality standards set by national
and international guidelines. However, the time between sampling and results takes at least one day. This delay in
results and interval between measurements makes it difficult to use the laboratory measurements for real-time
control of a treatment plant (van de Ven et al., 2010). In addition, it should not be underestimated that erroneous
control and measurement devices can also cause disturbances (van Schagen et al., 2010).
Retrieving reliable and robust on-line information is therefore important in order to be able to control a WTP. This
information can be retrieved from on-line sensors that measure a specific parameter directly, but also from generic
sensors that give indirect information. Roccaro et al. (2008), Rieger et al. (2004) and van den Broeke et al. (2008)
showed the ability of UV-Vis spectra measurements, measuring the absorbance of ultraviolet or visible light, to
estimate different parameters such as chlorine decay, nitrite and nitrate, ozone and assimilable organic carbon
(AOC) concentrations. These estimations were derived from algorithms developed, based on a change in UV-Vis
absorbance during a treatment step and laboratory measurements, using principal component analysis followed by
partial least squares regression. These types of generic sensors are so-called soft-sensors, sensors that require
software to give the required information. Juntunen et al. (2013) developed a soft-sensor to predict the turbidity in
treated water and to find the most significant variables affecting turbidity.
A soft-sensor can be developed in different ways, based on black box, grey box or white box modeling. The black
box approach is characterized by an empirical relation between the input and output. The relations are derived
from historical, full-scale plant, data. Thus, such a soft-sensor can only be applied in the situation where it has
been developed for, since a black box model is not valid when a process is operated outside the boundaries of
calibration (Kano and Nakagawa, 2008). Because the operation of a WTP is relatively constant, the calibration
dataset is normally rather limited, hampering the application of black box modeling. Grey box models are a
combination of black box models and white box models, such that it contains some more insight into the system
through the white box model, while still some parts of the model are data driven (Zyngier et al., 2001). White box
models mathematically describe the physical-chemical processes that take place in the treatment process.
Developing these models is time consuming, however, when developed, the process knowledge on the processes
are captured, leading to more generically applicable models (van der Helm and Rietveld, 2002).
Optimized control can only be reached if there is a high probability of detecting a bias in the operation of the WTP.
At the same time, from an economical perspective, the data should be obtained with minimal measuring efforts
and costs. Understanding the requirements with respect to on-line monitoring and data reliability is a first step
towards direct control of the drinking water production based on the incoming water quality. Therefore, in this
paper a design methodology is described on how to come to an optimized water quality monitoring scheme to
support direct control. This will be explained by means of a case study for a WTP.
**MATERIALS AND METHODS**
**Design methodology**
Van Schagen et al. (2010) developed a methodology for the design of a control system for drinking water treatment
plants. This methodology was based on experiences with control design procedures for chemical plants and was
modified to fit the main objectives of a drinking water treatment plant. In the basis, the same methodology was
used for the design of an optimized water quality monitoring scheme. The methodology takes into consideration
1) the objectives, 2) operational constraints and 3) disturbances. These first three steps determine the required
water quality parameters. The subsequent steps help to determine the conditions the water quality information
should comply with:
1. Determine treatment step objectives;
2. Determine operational control options;
3. Determine water quality parameters taking into consideration both process and control aspects;



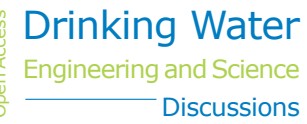

4. Identify process characteristics;
5. Evaluate available (indirect) measurements;
6. Determine individual monitoring strategy per treatment step.
7. Determine integrated monitoring strategy of treatment plant.

***Treatment step objectives***

The treatment step objectives depend on the feed water quality and the type of treatment step considered. The overall objective of a drinking water treatment plant is the production of safe drinking water fulfilling the quality standards defined by national and international guidelines. The main objective of a treatment step for an existing plant should be the focus on water quality and less on the chemical or energy consumption (van der Helm et al., 2008b). Therefore it should be evaluated which parameters, present in the feed water quality, can be influenced per treatment step. In order to do so process knowledge on the different treatment steps is indispensable (Poch et al., 2004). Van Schagen (2009) indicated that mathematical models are a powerful tool to evaluate the sensitivity to process objectives and disturbances and help find the appropriate controlled variables.

***Operational control options***

Depending on the design of the treatment step certain operational control options are available to make changes to the treatment process. Examples of operational control options are the change in chemical dosage, flow division and backwash and regeneration frequency. The primary focus is on the operational changes that can be performed within the existing plant lay-out.

***Required water quality parameters***

Based on the treatment step objectives and existing operational control options, the water quality parameters that are influenced by the treatment step are determined. Ideally these water quality parameters should be monitored. Besides the water quality parameters that are influenced by a treatment step, there are water quality parameters that influence the efficiency of a treatment step. For example, the water temperature has an effect on the ozone decay rate. The decay rate increases with increasing temperatures (Elovitz et al., 2000). This may result in a higher required ozone dose in summer time, taking into consideration that the disinfection requirements are also different with different temperatures.

***Process characteristics***

The required monitoring frequency and sensitivity of the selected water quality parameters may also vary depending on the process characteristics. The process characteristics describe the time interval during which changes occur and the order of magnitude in which changes occur. For instance, the contact time in an ozone reactor can vary from a couple of minutes to one hour, depending on the dimensions, while the time between two regeneration cycles of activated carbon typically is expressed in years. These different reaction times require different measurement frequencies. The order of magnitude relates to the required accuracy of the measurement. For example, ozone typically degrades quickly in water due to the reaction with organic compounds in the water. This determines that the required measurement sensitivity and accuracy should be high.

***Evaluate available measurements for the identified water quality parameters***

Based on the evaluation of the required water quality parameters and existing process characteristics the available (on-line) measurements should be evaluated. A wide range of measurements exist for determining water quality parameters, from certified laboratory measurements to on-line measurements. Depending on the variability of the process, the turnaround time of laboratory measurements is not always fast enough. To come to an optimal water quality monitoring scheme also on-line water quality sensors should be considered. In this study the following evaluation criteria for the available on-line sensors were assessed:
Easiness; is the sensor easy to use, is the measuring principle easy to understand;
Sensitivity; is the measurement range sensitive enough;
Maintenance; does the sensor require much maintenance;





Costs for laboratory measurements as well as the purchasing and maintenance costs for on-line sensors were
indicated. Besides on-line sensors developed to measure one specific parameter, available surrogate sensors, used
to estimate a water quality parameter value, and soft-sensors were assessed.
***Determine individual monitoring strategy per treatment step***
The individual monitoring strategy defines which water quality parameters per treatment step should be monitored,
with a selected frequency and location. The evaluation, of available measurements for the identified water quality
parameters forms the basis for the monitoring strategy, subsequently ranked by the most critical parameters in the
treatment plant. Criticality is determined by two factors, 1) parameters of which the measured concentrations are
close to the not to exceed limit and 2) parameters that can be potentially harmful to human health.
***Determine integrated monitoring strategy of treatment plant***
The integrated monitoring strategy defines which water quality parameters are monitored, taking into
consideration the interaction between the different individual treatment processes. The evaluation, of available
measurements for the identified water quality parameters forms the basis for the monitoring strategy, again ranked
by the most critical parameters in the treatment plant. The monitoring strategy can be imbedded into the process
control strategy to ensure optimized control based on the most critical parameters.
**Case study: Ozonation and biological activated carbon filtration at Waternet**
At the production location Weesperkarspel of Waternet, the water cycle company of Amsterdam and surroundings,
ozonation, pellet softening, biological activated carbon (BAC) filtration and slow sand filtration are the main steps
in the production of safe drinking water. The feed water is humics rich seepage water from the Bethune polder,
which is pre-treated by coagulation, sedimentation, approximately 100 days retention in a lake reservoir followed
by rapid sand filtration, before it is transported to the Weesperkarspel treatment plant. At Weepserkarspel, the
production of drinking water is roughly divided into two parallel lanes, each consisting of several individual
reactors/filters per treatment step. The control actions can be modified at individual level, however, for the purpose
of this Chapter it has been chosen to focus on the mixed influent and effluent only and not on the individual
reactor/filter level. The treatment processes ozonation and BAC filtration have been evaluated. These processes
are frequently applied at surface WTPs and are susceptible to changes in the feed water quality. Besides, these
processes have several control options and an interaction between the two processes exists.
Previously, the following on-line measurements were installed to monitor the ozonation and BAC filtration process
(Figure 1).

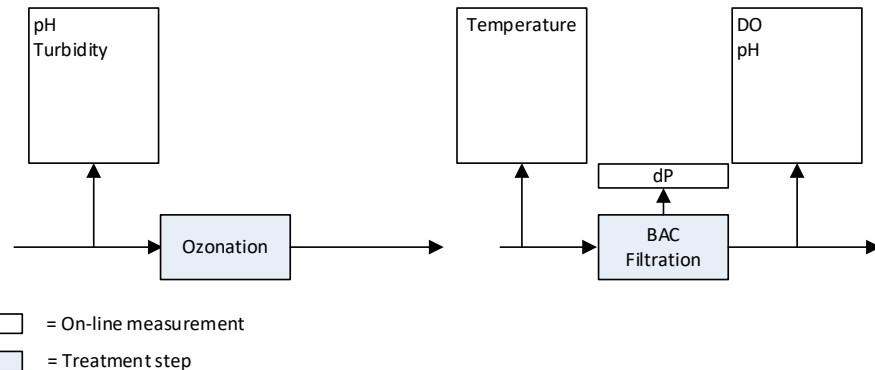

**Figure 1 Previous installed on-line measurements ozonation and BAC filtration at Weesperkarspel treatment plant**



IpH and turbidity were monitored at the influent of the ozonation step. The temperature was monitored in the influent of the BAC filtration. After BAC filtration dissolved oxygen (DO) and pH were measured, and the pressure drop was recorded over each of the individual BAC filters.

## RESULTS

The results of the evaluation of each step, to come to an optimised water quality monitoring scheme, are described below, followed by a discussion on the outcomes of the assessment versus the previous and current monitoring strategy. Research carried out at the pilot plant of Weesperkarspel was used to obtain full understanding of the processes taking place and enabling the determination of the objectives and required water quality parameters.

### Treatment plant objectives

In general the primary objective of ozonation is disinfection (von Gunten, 2003b). Besides, ozonation is frequently used for the oxidation of organic micro pollutants, taste, odour and colour producing products and natural organic matter (NOM), transforming higher molecular weight compounds into lower molecular weight compounds. For the ozonation step at Weesperkarspel, the specific objectives are disinfection and oxidation of NOM (van der Helm, 2007).

The general objective of activated carbon is the removal of organic micropollutants, removal of precursors of disinfection by-products and the removal of organic compounds causing colour, taste and odour issues (van der Aa et al., 2011). When activated carbon is preceded by a pre-oxidation step, the biological activity in the water and on the activated carbon is enhanced, resulting in BAC filtration. At the same time ozonation increases the polarity, resulting in a decrease in adsorption capacity (Sontheimer et al., 1988). As a result, NOM is removed through both biodegradation and adsorption. At Weesperkarspel the purpose of BAC filtration is the removal of organic matter, to prevent biological growth in the distribution system and to remove toxicity, taste and odour causing compounds (Graveland, 1996). Besides, the BAC filters remove the carry-over from the preceding pellet softening step.

### Operational control options

The production flow is controlled by the demand for drinking water. The buffering capacity in the treatment plant is the clean water storage reservoirs situated before the water is distributed to the customers (van Schagen et al., 2010). To ensure sufficient reliability, the treatment plant is set up in a redundant way with multiple lanes operated in parallel. It is possible to change the flow division over the different production lanes, however this is only done when one of the lanes has less treatment capacity or is out of production due to e.g. maintenance. Therefore, in this case, production flow was not considered as a control action.

The only remaining control action for ozonation is the ozone dosage. The ozone dosage is obtained by a combination of ozone in gas concentration and the gas flow. Both parameters can be adjusted to obtain the desired ozone dosage.

For BAC filtration the control actions within the existing treatment setup are the backwash frequency, currently operated at every couple of days till once a month interval per filter and backwash program, currently a combination of air and water is used. The activated carbon is regenerated every year to three years. Carbon dioxide is dosed before the BAC filters to correct for any high pH resulting from the caustic soda dosage in the pellet softeners. This control action is thus more related to the functioning of the pellet softeners and therefore not included in the overview provided in Figure 1. A high pH could negatively affect the biodegradation efficiency (Seredyńska-Sobecka et al., 2006) and promotes precipitation of calcium carbonate on the activated carbon grains. Oxygen and caustic soda can be dosed in the effluent of the BAC filters to correct low pH and oxygen concentrations as a result of the biological activity in the filters.

### Required water quality parameters



As indicated previously, ozone is an unstable oxidant in water. Ozone decomposition in water consists of a fast initial phase (seconds range) and second phase (minutes range) during which ozone concentration decreases via first order kinetics and disinfection of the more resistant pathogenic microorganisms takes place (von Gunten, 2003a;van der Helm et al., 2008a). A commonly used method to determine the disinfection capacity of ozonation is by calculating the exposure of pathogens to ozone, expressed as the Ct value, a product of the (residual) concentration of the disinfectant (C), in this case ozone and contact time (t) (WHO, 2008).

Water quality parameters that influence the efficiency of the ozonation step are temperature, pH and, for Weesperkarspel relevant, scavengers such as NOM concentration and character (von Gunten, 2003a). A measurement commonly used to indicate the NOM concentration is the dissolved organic carbon (DOC) concentration. The DOC concentration is determined by filtering the sample over a 0.45 µm filter and measuring the total organic carbon (TOC) concentration. In order to assess the character of NOM, the specific UV absorbance (SUVA) can be calculated by dividing the UV absorbance measured at a wavelength of 254 nm (UV254) by the DOC concentration (van der Helm et al., 2008b;Edzwald and Tobiason, 1999). These water quality parameters play a role in the ozone dosage required to achieve the desired disinfection and should therefore be monitored. For Weesperkarspel it was determined that disinfection of Giardia, Cryptosporidium and Campylobacter is sufficient to determine the microbiological safety of the water (van der Helm et al., 2008b). To be able to monitor the efficiency of the ozonation step, at least one of the following parameters should be measured:
- Pathogenic mirco-organisms such as Cryptosporidium, Giardia and Campylobacter.
- Ozone concentration at different contact times, to be able to determine the Ct value (van der Helm et al., 2009);

During ozonation disinfection by-products are formed. The oxidation of NOM promotes the presence of AOC concentration in water (van der Kooij et al., 1989). AOC promotes regrowth of bacteria in a distribution system, amongst others, and, therefore, should be sufficiently removed in the subsequent treatment steps. Water without residual chlorine is considered to be biologically stable if the AOC concentration is below 10 µg Acetate-C/L, whereas water with residual chlorine is defined as biologically stable for AOC concentrations below 50 µg Acetate-C/L (van der Kooij, 1992;Escobar et al., 2001). Besides AOC, bromate is formed if bromide is present in the feed water. Bromate is possibly (IARC, 1999) or probable (USEPA, 2018) carcinogenic to humans.

During BAC filtration, biodegradation takes place by microorganisms, present on the external surface and in the macro-pores of the BAC filter grains, that biodegrade the NOM in the water (Servais et al., 1994). The activity of the microorganisms (biomass) determines the degradation rate of NOM (Lazarova and Manem, 1995). The activity and concentration of the biomass depends on the concentration of nutrients (carbon, phosphate and nitrogen), the dissolved oxygen concentration, temperature, pH and residual disinfectant in the feed water (Simpson, 2008). Uhl and Gimbel (2000) described that for the biological removal of ammonia, the deposit of bacterial cells from the influent was necessary to maintain a solid biofilm. However for Weesperkarspel it was shown that the feed in bacterial cells to the BAC filters was not necessary to obtain a sufficient biodegradation efficiency (Ross et al., 2019), hence no on-line measurement of ATP or flowcytometry was required. Besides biodegradation taking place, adsorption of NOM and toxic, colour, taste and odour compounds takes place. In addition, at Weesperkarspel, BAC filtration is simultaneously applied for the removal of suspended solids and carry-over. Due to clogging of the filter bed by suspended solids, carry-over and in some cases biomass, the filters need to be backwashed frequently. The pressure drop over the filters and turbidity in the effluent indicates the state the filter is in, and whether it needs to be backwashed. In case of Weesperkarspel the pressure drop is the determining parameter.

*Process characteristics*
Ozone is dosed to the water, after which reaction takes place in the seconds to minutes range. A change in ozone dose or change in feed water quality can have an immediate effect on the effluent quality. In the past, the dosing strategy was determined by the water temperature, with two different set points, below 12 ºC and above 12 ºC. Van der Helm et al. (2009) suggested that this negatively influenced the disinfection during ozonation. However, more detailed research by Wiersema (2018) could not confirm this. Since ozonation is one of the main processes that can achieve disinfection, high frequency monitoring is required enabling direct control of the ozonation step.



In contrast to ozonation, BAC filtration is not a dosing process, but a separation/degradation process by means of filtration, adsorption and biodegradation. The different processes all have their associated time intervals. The shortest time interval is the clogging of the filters, which, depending on the location in the treatment train, needs to be carried out every couple of days till once a month. Backwashing occurs based on pressure drop over the filter or after a maximum period of time. The pressure drop should be monitored on a regular basis.

As indicated in the required water quality parameters section, the activity of the biomass present on the carbon grains determines the biodegradation efficiency. Ross et al. (2019) showed that a change in feed water quality does not necessarily result in a change in effluent quality, hence there is no direct need for close monitoring of the filters. In case the feed water quality changes for a longer period of time, the biomass will adopt itself to the new situation, which can take up to 2-3 months (Servais et al., 1994).

Depending on the NOM loading, the activated carbon starts showing break-through of organic micro pollutants and pesticides after a run time of 6-9 months if no biodegradation takes place, while if biodegradation takes place this can last up to 2-5 years before the activated carbon needs to be regenerated (Simpson, 2008). Although BAC filters have proven their ability to intercept sudden changes in water quality, the DO can be used as an indicator for the biological activity in the filter and identifying any disruptions taking place (van Schagen, 2009).

### *Evaluate available measurements for the identified water quality parameters*

A summary of the required water quality parameters, as determined in the paragraphs describing the water quality parameters, can be found in the first column of Table 1 (ozonation) and Table 2 (BAC filtration). In the second column it is indicated per parameter if an on-line measurement, able to measure at the limit of detection required, is available. Depending on the monitoring frequency required, as described in the process characteristics paragraphs, it was determined if a parameter should be available on-line. If the monitoring frequency should be daily or more, it was indicated with a yes in the third column. To gain a better understanding of the applicability of the on-line sensors, the ease of use, sensitivity and maintenance requirements were evaluated in columns four through six. The costs related to a measurement in lab and installation of an on-line sensor are listed in column seven.

Evaluation of availability on-line sensors and its characteristics was based on literature research, indicated by the references included per parameter. Besides on-line sensors that measure one specific parameter, available related surrogate parameters (column eight) and soft-sensors (column nine) were also captured. It should be noted that for some surrogate parameters and soft-sensors a start concentration is required first before the concentration of the requested parameter can be estimated.



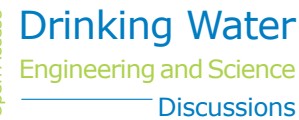

**Table 1 Summary water quality parameters required to monitor ozonation and associated available on-line sensors**

| Parameter | On-line available | On-line required | Easy | Sensitive enough | Maintenance | Costs lab/online | Surrogate parameters | Soft-sensor available |
|---|---|---|---|---|---|---|---|---|
| pH | Yes (Banna et al., 2014) | Yes | Yes | Yes | Moderate, needs regular calibration | lab/online: low | No | Yes through water quality (WQ) modeling after dosages of a base or acid based on measured influent pH (van Schagen et al., 2009) |
| Temperature | Yes (Banna et al., 2014) | Yes | Yes | Yes | Low | lab/online: low | No | No |
| DOC | Yes via TOC measurement (Hall et al. 2007) | Yes | Moderate | Yes | High, 0.45 μm filters and reagents are required to be replaced | lab: moderate online: high | $UV_{254}$ or a $UV_{280}$, UV wavelength at 254 or 280 nm related to reactivity of the organic carbon with ozone (Westerhoff et al., 1999) | Yes, based on range of UV wavelengths (Langergraber et al., 2003) |
| $UV_{254}$ | Yes (Hach, 2018) | Yes | Yes | Yes | Yes | lab: low online: moderate | UV/Vis measurement, measuring all wavelengths between 200 – 735 nm | n.r. |
| Pathogenic micro-organisms | No | Yes | n.a. | n.a. | n.a. | lab: high online: n.a. | Ct value related to inactivation of Giardia after measuring influent concentration (USEPA, 1989) | Yes, Ct value estimation by means of WQ modeling (van der Helm et al., 2009) or algorithm based UV/Vis-spectra measurements after measuring influent concentration (Ross et al., 2016) |
| AOC | No | Yes | n.a. | n.a. | n.a. | lab: high online: n.a. | Yes (Hammes and Egli, 2005) | Yes, through WQ modeling by van der Helm et al. (2009) or algorithm based on UV/Vis-spectra measurements (Ross et al., 2016) |
| Bromate | No (ThermoFisher, 2018) | Yes | n.a. | n.a. | n.a. | lab: moderate online: n.a. | Yes, Ct value has linear relationship with bromate (van der Helm et al., 2008a) | Yes, through WQ modeling by van der Helm et al. (2009) or UV/Vis-spectra measurements (Ross et al., 2016) |
| Bromide | No | No | n.a. | n.a. | n.a. | lab: moderate online: n.a. | n.r. | n.r. |
| Ozone concentration in water | Yes (Hach, 2018) | Yes | Moderate | No | Moderate, regular cleaning required | lab/online: moderate | Yes, UV absorbance from 185-350 nm (Molina and Molina, 1986) | Yes, developed based on UV measurement (van den Broeke et al., 2008) |

n.a.= not applicable, n.r. = not required.



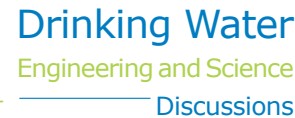

**Table 2 Summary water quality parameters required to monitor BAC filtration and associated available on-line sensors**

| Parameter | On-line available | On-line required | Easy | Sensitive enough | Maintenance | Costs lab/online | Surrogate parameters | Soft-sensor available |
|---|---|---|---|---|---|---|---|---|
| DO | Yes (Banna et al., 2014) | Yes | Yes | Yes | Low | lab/online: low | No | No |
| Phosphate | Yes (Schlegel and Baumann, 1996;Hach, 2018) | No | Yes | No | Moderate, reagents are required to be replaced | lab: moderate online: moderate | n.r. | n.r. |
| Nitrogen | No | No | n.a | n.a. | n.a | lab: moderate online: n.a. | n.r. | n.r. |
| DOC | Yes via TOC measurement (Hall et al. 2007) | No | Moderate | Yes | High, 0.45 μm filters and reagents are required to be replaced | lab: moderate online: high | n.r. | n.r. |
| AOC | No | No | n.a. | n.a. | n.a. | lab: high online: n.a. | n.r. | n.r. |
| Viable bacterial cells | Yes (Besmer et al., 2017) | No | Moderate | Yes | Moderate | lab: moderate online: high | n.r. | n.r. |
| pH | Yes (Banna et al., 2014) | Yes | Yes | Yes | Moderate, needs regular calibration | lab/online: low | No | Yes through water quality (WQ) modeling after dosages of a base or acid based on measured influent pH (van Schagen et al., 2009) |
| Temperature | Yes (Banna et al., 2014) | Yes | Yes | Yes | Low | lab/online: low | No | No |
| Pressure drop | Yes (van Schagen et al., 2008) | Yes | Yes | Yes | Low | lab: moderate online: low | n.r. | n.r. |

n.a.= not applicable, n.r. = not required.

*Determination of individual monitoring strategy per treatment step*
Figure 2 shows the individual monitoring strategy per treatment step determined by the water quality assessment
captured in Table 1 for ozonation and Table 2 for BAC filtration. The results are described in detail below.

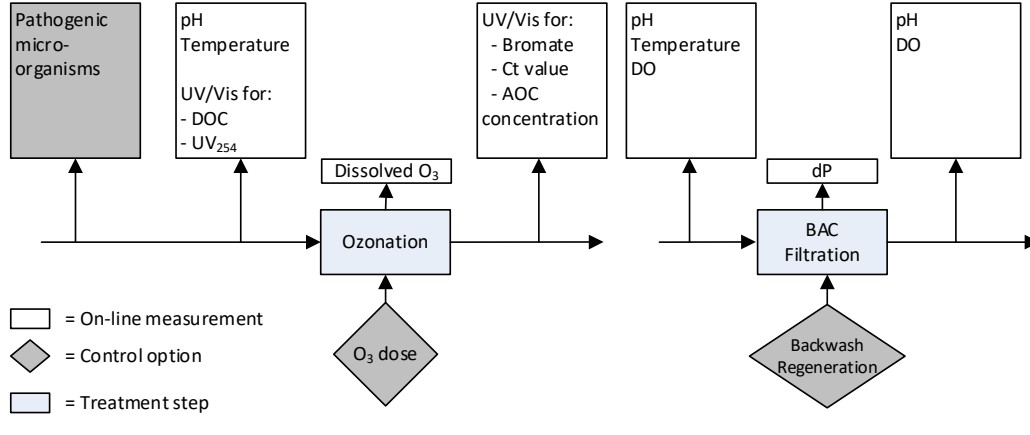

**Figure 2 Required on-line water quality information for optimized monitoring and control of ozonation and BAC**
**filtration**
**pH, temperature, and DO**
Being compliance parameters published by the WHO, there are sufficient on-line sensors available to measure the
pH, temperature, and DO. These sensors are relatively easy to use and sensitive enough. The pH sensor requires
frequent maintenance. The costs of measurement, either on-line or in laboratory are low. The efficiency of ozone
is, amongst others, determined by the pH and temperature and should therefore be monitored continuously. The
DO and pH are a continuously controlled effluent parameter in BAC filtration. The pressure drop indicates if a
filter needs to be backwashed. The DO and pH are an indicator for the biological activity in the filter and capable
of identifying any disruptions taking place (van Schagen, 2009).
**DOC and UV$_{254}$**
The NOM concentration, measured through DOC, is a scavenger and does directly interfere with disinfection,
requiring to be monitored in the influent of the ozone step. The used ozone dosages hardly affect the DOC
concentration, limiting the need for monitoring downstream of the ozone step (van der Helm et al., 2008a). For
TOC there is an on-line sensor available which measures sensitive enough. By inclusion of a 0.45 μm filtration
step the DOC is determined. It does require frequent maintenance for replacing the 0.45 μm filters and reagents.
The on-line sensors are still expensive whilst the lab measurements are cheap and around 20 euros per sample.
Alternatively, an UV absorbance sensor measuring the UV absorbance at wavelength of 254 or 280 nm can be
used as a generic sensor providing insights in the reactivity of ozone with the organic matter (Westerhoff et al.,
1999). Besides direct measurement or a generic sensor, Langergraber et al. (2003) developed a soft-sensor allowing
to estimate the DOC concentration based on measured UV/VIS wavelengths and by applying principal component
analysis followed by partial least squares regression. These soft-sensors do require to be calibrated locally based
on an obtained dataset from lab measurements. The UV/Vis sensor is, besides regular cleaning, easy to maintain,
and less than half the price of a specific TOC sensor. Besides DOC, UV254 also determines the efficiency of ozone
and should therefore be monitored continuously. A specific on-line sensor is available which only measures
UV254, is easy to use, sensitive and low in maintenance and costs. An alternative generic sensor is the UV/Vis
sensor which measures all wavelengths between 200-735 nm. This should only be used instead if the sensor is
used to measure other parameters, such as DOC, as well.

## AOC, bromate and bromide

AOC and bromate are disinfection by-products formed during ozonation. Depending on the influent concentrations of DOC and bromide and the amount of ozone dosed, the AOC and bromate concentration are determined. There is no on-line sensor available for measuring the AOC concentration in accordance with the approved standard methods (Eaton et al., 2005). AOC is one of the disinfection by-products that needs to be monitored. A change in organic matter composition and/or ozone dose will directly result in a change in AOC concentration, therefore requiring on-line monitoring in the effluent of the ozone step. AOC is subsequently biodegraded in BAC filtration step and enhances the microbiological activity in the filters. Ross et al. (2019) showed that a sudden change in AOC concentration does not result in a direct deterioration of the effluent quality of the BAC filters. Therefore, a continuous monitoring of the AOC concentration in the effluent of the BAC filter is not required. The lab measurements are high in costs, due to the labour intensity of the analysis. Hammes and Egli (2005) developed a quicker laboratory method to determine the AOC concentration using flow cytometry. Until now this method is only available as off-line method and therefore not suitable for on-line monitoring. The water quality model developed by van der Helm et al. (2009) is able to predict the formation of disinfection by-products such as AOC by using Matlab/Simulink®. Another soft-sensor is the software algorithm published by Ross et al. (2016) that uses different UV/Vis wavelengths to predict the AOC formation.

There are no on-line sensors available for measuring the bromate and bromide concentration. Bromate needs to be monitored for compliance since it is possibly carcinogenic and is not removed in existing downstream treatment steps. A change in bromide concentration or a change ozone dose can impact the bromate concentration directly. The bromide levels in the influent of the Weesperkarspel treatment plant have been very stable, requiring no need for continuous monitoring. Since the bromate levels can change with changing ozone dose, on-line monitoring of bromate in the effluent of the ozone step is proposed. The lab measurements are moderate in costs, due to the reagents required. Van der Helm et. al. (2008a) found a linear relationship between the bromate concentration and Ct value, allowing the Ct value to be a surrogate parameter once the initial bromate concentration is known. Cromphout et al. (2013), found a linear relationship between ozone dose, temperature and bromate formation. These models can be used to predict the bromate concentration based on the ozone dosed, temperature, pH and bromide concentration in the influent. Another available soft-sensor is the software algorithm published by Ross et al. (2016) using different UV/Vis wavelengths to determine the Ct value and bromate formation. It should be tested till what extent these algorithms can be locally calibrated for changing bromide concentrations.

## Pathogenic micro-organisms and ozone concentration in water

There are no on-line sensors available to specifically measure a certain pathogenic microorganism. The lab measurements are high in costs, due to labour intensity of the analysis. The pathogenic microorganism concentration in the influent together with above parameters do determine the required ozone dosage and therefore require continuous monitoring. The USEPA (1989) published Ct values for determining the log inactivation of pathogenic microorganisms for different water temperatures. This allows the Ct value to be used as a surrogate parameter if the influent concentration is known. The water quality model developed by van der Helm et al. (2009) is able to predict the Ct value based on above measured parameters and applied ozone dose. In addition, Ross et al. (2016) published a software algorithm that uses different UV/Vis wavelengths to determine the Ct value. Verification via lab analysis of pathogenic microorganisms on a weekly/monthly basis, depending on the variability of the source water quality, will help determine the log inactivation and associated Ct value to be achieved. Besides using soft-sensors to determine the Ct value based on a change in UV/Vis pattern, the ozone in water can be determined by on-line measurements. These measurements do require local calibration by means of lab measurements. It is an easy and sensitive measurements that does require regular maintenance to prevent biofouling. Cost of on-line and lab measurements are moderate due to the calibration fluid required. In order to be able to determine the Ct value based on the ozone in water concentrations, multiple sampling points are required in space.

## Phosphate and nitrogen

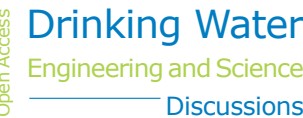
Phosphate, nitrogen and carbon are the nutrients required for the microbiology in the BAC filters to grown on.
Phosphate is a frequently on-line measured and controlled parameter in wastewater environments. The available
on-line measurements are easy to use, sensitive enough, but do require regular maintenance due to reaction agents
used. The costs of both lab and on-line application are moderate. To the authors knowledge there are no on-line
nitrogen measurements available. The costs of lab measurements are moderate. In the current treatment plant setup
there is no option to alter the phosphate or nitrogen concentration (by means of dosing) and as a result there is no
need to continuously monitor these concentrations in the influent of the BAC filters.
**Viable bacterial cells**
Viable bacterial cells are present in the surface water. During ozonation typically disinfection of viable bacterial
cells takes place, which subsequently can regrow in following treatment steps (Vital et al., 2012). The
determination of viable bacterial cells has developed in the last couple of years from a laborious intensive
measurement using microscopy, to rapid determination in the lab using flow cytometry to customizing the flow
cytometry equipment for on-line applications (Besmer et al., 2014;Besmer et al., 2017). Ross et al. (2019) showed
that the effect of viable bacterial cells in the influent of the BAC filters is limited in respect to the performance of
the BAC filters, therefore discarding the need for on-line monitoring. The costs of both lab and on-line
measurements are still high but expected to reduce in future as per the innovation taking place to enhance rapid
detection.
**Pressure drop**
The pressure drop is typically measured to determine the clogging ratio in the filter bed. Pressure drop
measurements are available on-line and have been fully developed. It is an easy measurement, which is sensitive
and low in maintenance. The costs are low. For BAC filtration it is, besides turbidity, the main indicator if a filter
is clogging and needs backwashing. On-line monitoring is therefore required and frequently applied.
***Determination of integrated monitoring strategy of treatment plant***
When evaluating the ozonation and BAC filtration step as an integrated system, it is not required to monitor the
AOC in the effluent of the ozonation due to the robustness of the BAC filtration step (Ross et al., 2019). The DO
concentration in the influent of the BAC filter will always be sufficient as a result of the preceding ozonation step,
therefore there is no need to continuously monitor this concentration in the influent. For Weesperkarspel, the
temperature of the water and pH will not change due to application of ozonation, hence there is no need to monitor
this in the influent of the BAC filters.
In Figure 3 the current monitoring strategy of Weesperkaspel is shown. This strategy was adjusted per the
outcomes of the different research described in this paper (van der Helm, 2007;Ross et al., 2016;van Schagen,
36 2009).

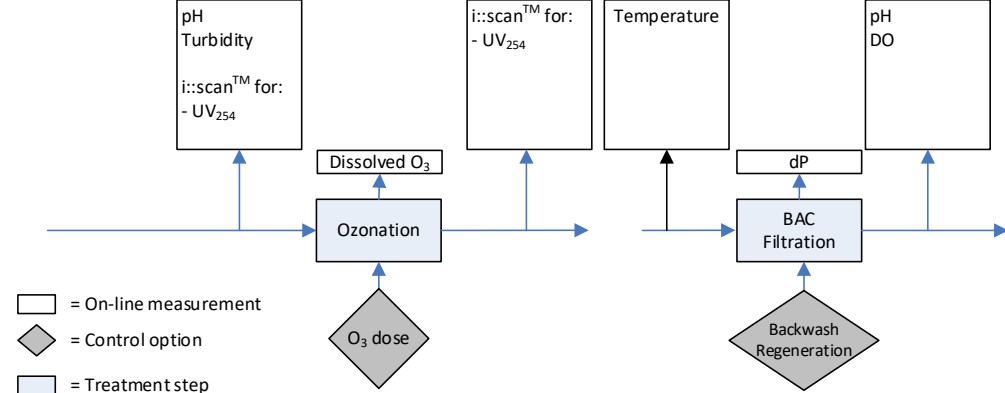



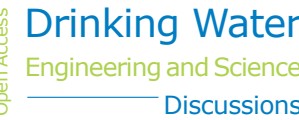

**Figure 3 Current on-line water quality monitoring of ozonation and BAC filtration at Weesperkarspel treatment plant**

When comparing the sensors installed in Figure 3 with Figure 2, considering the sensors that can be skipped based on the integrated approach, only 4 differences are observed. In the influent of the ozone step only UV254 is measured instead of UV254 and DOC and the turbidity is measured. In the effluent of the ozone step the i::scanTM is installed measuring at a wavelength of 254 nm instead of the s::scanTM able to measure the full spectrum allowing for estimation of bromate and Ct value. However, the Ct value can also be calculated by the installed ozone measurements and the UV254 can give a good indication of the achieved Ct as well (Westerhoff et al., 1999). No differences are observed for the BAC filtration step, when considering the integrated approach.

**DISCUSSION**

*Advances in on-line water quality monitoring*
Evaluation of available on-line sensors showed that the parameters typically measured to show compliance with the WHO standards are commonly available (Adu-Manu et al., 2017). Direct measurements of the more complex parameters such as AOC and bromate are not available on-line. When looking at required on-line information for integrated control of ozonation and BAC filtration, bromate is to be monitored continuously. In this case the use of soft-sensors, able to estimate the bromate and AOC formation, help to gain continuous on-line data. Besides using soft-sensors as surrogate sensors for parameters currently not available on-line, they can also provide a cost effective alternative when used to determine multiple parameters required through one single instrument. Examples in this case were the use of UV-Vis sensors for the determination of UV254 concentration in the influent, the estimation of DOC in influent and effluent, formation of bromate and AOC during ozonation and estimation of Ct value in the effluent of the ozonation step.

*Reliability of the data*
On-line identification of disturbances is only possible if the identified water quality data are accurate and continuous (van Schagen et al., 2010). Furthermore the confidence the operators have in the data is crucial, especially when soft-sensors are applied instead of direct measurement (Ikonen et al., 2017). If possible, measurement via two different methods can be applied for a period of time, to gain confidence by the operators to rely on soft-sensors to provide with the correct information. In this case the Ct value can be obtained via ozone in water measurement multiplied by contact time or estimated via the change in UV-Vis measurement. It should be recognized that the use of on-line sensors does require knowledge of the use of the sensors and (frequent) maintenance to ensure the reliability of the data.

*Direct control based on water quality*
When comparing the previous on-line information (Figure 1) with the current on-line sensors placed at Weesperkarspel (Figure 3) it can be seen that in the current situation more on-line information is available. The current situation comes close to the required situation as depicted in Figure 2, when considering the integrated approach. Currently the installed sensors act as an early warning system to flag any deviations in water quality and operation. The next step would be the direct control based on water quality.

Fluctuations in incoming water quality and subsequent required change in ozone dose to achieve the objectives set forth of achieving sufficient disinfection while minimizing the disinfection by-product formation require direct continuous monitoring and direct control. Van der Helm et al. (2009) suggested that the control of ozonation step, and balancing between disinfection and disinfection by-product formation, can already be greatly enhanced when adjusting the ozone dose based on the measured water temperature. However, more detailed research by Wiersema (2018) could not confirm this. By adjusting the ozone dose on incoming NOM concentration, the balance between disinfection and by-product formation might be improved.

**CONCLUSIONS**





The main objective of this paper was to develop a design methodology able to determine an optimised water quality monitoring strategy to support future direct control of the drinking water treatment plant based on incoming water quality. A seven step approach was defined, and each step was demonstrated for the treatment processes ozone and BAC filtration. It was shown how the previous on-line water quality monitoring program of the treatment plant Weesperkarspel was optimised and subsequently can be finetuned in future.

Evaluation of available on-line sensors showed that the parameters typically measured to show compliance with the WHO standards are commonly available. Direct measurements of the more complex parameters such as AOC and bromate are not available on-line. The use of soft-sensors, able to estimate the bromate and AOC formation, help to gain continuous on-line data. Besides using soft-sensors as surrogate sensors for parameters currently not available on-line, they can also provide a cost effective alternative when used to determine multiple parameters required through one single instrument. Examples in this case were the use of UV-Vis sensors for the determination of UV254 concentration in the influent, the estimation of DOC in influent and effluent, formation of bromate and AOC during ozonation and estimation of Ct value in the effluent of the ozonation step. The on-line data obtained by the (soft-) sensors will help the operator to control the treatment plant based on its objectives and provide continuous information whether the processes are operating within the required operational window.

## ACKNOWLEDGEMENTS

This research was financially supported by the 6[th] EU framework project TECHNEAU (contract number 018320).

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
