# Peer review of "Design methodology to determine the water quality monitoring strategy of a surface water treatment plant in the Netherlands"

_Drinking Water Engineering and Science, 2019_

## Referee Comment (RC1) · Joep van den Broeke (Referee) · 18 Jun 2019

The paper "Design methodology to determine water quality monitoring strategy of surface water treatment plants" describes the application of a well-considered generic design strategy to analyse the possibilities and impossibilities of using available water quality sensors at one particular water treatment plant.

The design strategy has been derived from a similar strategy developed for a control-design methodology by the (co)authors van Schagen en Rietveld (van Schagen et al in Water Science and Technology, 10, 121-127, 2010).

[Figure]

After a discussion of the aspects to be considered in each of the steps of assessment proposed, one case study is discussed in which an existing monitoring strategy was analysed and optimised.

Unfortunately, it is in the description of this case study and the subsequent conclusions, that the paper fails to address a number of essential elements that are necessary to judge the effectiveness of the design methodology.

Whereas sensor technology, characteristics of available types of instruments as well as treatment process characteristics and operational handles are described in substantial detail, no attention is giving to the following aspects:

1) although making selection a combination of sensors, and suggesting (page 13, lines 37-38) that this new combination was actually installed, the only comment the authors make about the results achieved is that more information is now available (page 13 line 38) and that the balance between disinfection and by-product formation might be improved (page 13 line 49). The former is not an achievement of the presented methodology (any additional sensor installed would have resulted in more information becoming available) and the latter also follows from previous works by the authors, and but this is not confirmed in this study. Therefore, no improvement in process control or effluent water quality (the goals of this study, page 2, lines 31 - 37) are demonstrated.

2) As the authors present a method to determine (and / or optimise) a water quality monitoring strategy, it would be logical that the paper presents the results of the selected strategy and demonstrates that the outcome of the methodology actually produces an optimal monitoring setup. The authors do describe the selection process by describing both sensor options and processes (but not presenting any new information) but do not present any cost benefit analysis of the selected strategy, nor do they give any proof of how it improves the pre-existing monitoring setup at the location analysed. Therefore, the effectiveness of the design methodology cannot verified based on the results presented.

3) An implementation section, describing the experience with the actual deployment and operation of the sensors, this validation the methodology is missing. For an example, this reviewer would like to point the authors to the van Schagen paper (2010) from which the method described here was described. This paper has an implementation section where the approach is validated

As result of the above the authors' conclusion (also found in the abstract) that the water quality programme of the Weesperkarspel plant was optimised is not sufficiently supported.

The same is true for the conclusion on the use of soft-sensors (also mentioned in the abstract) - the authors state that the use of soft sensors helps to gain online information on parameters for which no online sensors are available. Although this may be correct, no new evidence to support this is presented in this paper. For this to be presented as one of the two major conclusions of the paper, it would be reasonable to expect the authors performed and analysis on the effectiveness of soft sensors in this type of application, and / or would have performed a cost benefit analysis (including CAPEX as well as OPEX costs) of using soft sensors vs. direct measurements.

Specific comments

Page 1 - line 26: The authors appear to extrapolate the situation in the Netherlands to a generic statement (regarding sampling frequency). This is, however, incorrect. The sampling frequency depends on the size of the utility and the population served. It also depends on the parameter analysed. This statement currently is too generic. Either it should be restricted to the situation in the Netherlands (where all water companies / treatment plants are so large daily sampling is required or the water companies themselves take daily samples even if not required), or this statement should be rephrased taking into account daily practice also at smaller utilities in other countries.

Page 2 - line 3: time between sampling and analysis takes at least one day. The authors ignore the intermediate option of rapid tests that can be performed on-site.

Utilities use such rapid tests for on-site measurements, e.g. to calibrate online sensors but also for data collection. Various parameters can even only be measured in this way (e.g. DO, O3, ...). It might be worth adding this type of analysis to the discussion, especially in view of the comment in the next sentence - such rapid tests can be used to verify performance.

Page 2 - lines 5 - 6: In addition, it should not be underestimated that erroneous control and measurement devices can also cause disturbances (van Schagen et al., 2010).

It is not clear what the authors are trying to say with this sentence. Is the goal of this sentence to indicate that human operators can make mistakes and online sensors will detect this? Then remove the reference to erroneous measurement devices. If the goal is to stress the importance of correct use (installation, maintenance, ...) of online sensors, then this should be discussed separately. Currently, this sentence suggests online sensors are not trustworthy, whereas the next paragraph stresses the usefulness of such devices. These are now contradicting each other.

Page 3 - line 39: please reconsider this statement; fast degradation does not necessarily mean a need for high sensitivity and high accuracy. Sensitivity and accuracy will depend on where the measurements are taken and what the purpose of the measurement is. Examples: Does one want to verify the concentration dosed (at injection point) then the decay is less important and concentration will still be high. Downstream concentrations will be lower. but if initial concentration were high (e.g. with other chemical than ozone) still a not-so-sensitive sensor would still do the job. It seems that a fast measurement is most important for rapidly decaying ozone.

Page 4 - lines 26 - 28 Here it is stated the chapter focuses on mixed influent and mixed effluent for the processes ozonation and BAC filtration. However, in the preceding sentence the authors described the treatment plant consists of two parallel lanes. If this is correct, then the methodology appears to be incorrectly applied as it focused on the non-existent situation of mixed ozonation effluent and mixed BAC influent. Could the

authors clarify this and if this is indeed the case (2 treatment lanes and no combination of the water between ozonation and BAC filtration), please correct the text. This can be solved by rephrasing lines 26-28 and removing the reference to the mixed streams.

Page 5 - lines 21 - 22: The authors state that the adsorption capacity (of the carbon filters) decreases with increasing polarity (of the organic matter). It appears that the equilibrium between adsorption and desorption will shift and the affinity of more polar organic matter for the carbon filter surface will be reduced. However, the adsorption capacity of the carbon (the maximum amount of material it can adsorb) should not be affected, as the ozonation does not change the surface properties of the adsorbent.

Page 5 - lines 22 - 23: As a result NOM is removed through ... adsorption. The current wording states that because NOM is oxidised and becomes more polar it is removed through biodegradation and adsorption. Is this true, i.e. would NOM be removed only through biodegradation or adsorption if no ozonation takes place?

Page 5 - lines 33 - 34: Therefore, ... control action. This appears to be a missed opportunity. The authors indicate the flow through the plant is variable (because determined by the demand for drinking water). This means the flow through the treatment lanes is variable. Keeping flow constant could be a very useful control measure for the performance of individual treatment steps. Changes the division of the water over the different production lines, (although not used currently) could thus be used to manage the performance of the treatment processes, and optimise it. E.g. though keeping one lanes at ideal operational conditions, which might result in better net quality of the mixed effluent. It would have been worth investigating this possibility. Eliminating this option at the beginning of the evaluation was a missed opportunity. Did the authors consider this type of evaluation?

Have the authors considered this possibility? Is there information (other than the fact that the plant operators do not use this control option) that it would not be been relevant?

Page 5 - lines 45-46. Because pH is important for the functioning of the BAC filters, can one be sure that the influent water (the effluent of the pellet softeners) is always at the correct / optimal pH? If this is the case, it is correct to disregard this parameter for control of the BAC. If the pH can be variable (e.g. because there is no real-time control of the CO2 dosing), then pH control should be seen as a control action for the BAC, even though the actual dosing equipment might be part of the pellet softener. Please add a statement that explains why it is not necessary to adjust pH before BAC, e.g. because it is always the same or because it is actively controlled in the softening.

Page 6 - lines 14 - 15: Monitoring of TOC/DOC/SUVA will only be necessary if the composition of the NOM changes. The authors describe the water is seepage water, which is groundwater. Please explain why this variable is relevant for the operation of this treatment plant: is there a variable composition in NOM that would necessitate monitoring in this example.

Page 6 - line 28: Bromate is possibly or probable carcinogenic. Suggest to refer to only one reference that reflects the latest insights into this topic. Referring to both possibly and probable is confusing.

Page 6 - line 38: ATP or flowcytometry. Please explain this statement. There was no reference before to these methods and as to why they would be required. Therefore, the statement on the absence of a need for these methods is confusing and seems irrelevant.

Page 8 - bromide: bromide sensors (ion selective electrodes) are available from various manufacturers. Please correct.

Page 8, Ozone, soft sensor available. Yes, developed based on UV measurement The measurement of ozone using UV spectroscopy is a direct measurement; O3 has a distinct spectrum which is directly measured. In the case of a soft sensor a number of measurements are taken together to estimate a parameter which can not directly be measured. For O3 by UV spectroscopy this is explicitly not the case, as the O3

spectrum itself is directly measured.

Page 12 - lines 27-28: When evaluating... it is not required to monitor ... due to robustness of the BAC filtration step. As the fact that it is not necessary to monitor the AOC was already known from previous work by the authors, why did they ignore this knowledge in the preceding discussion. This type of prior knowledge should flow into the design methodology as early as possible, as it prevents a waste of time (e.g. in this case the discussion on AOC could have been skipped).   Technical corrections Page 3 line 43: in "A wide range of measurement" Methods or Methodologies or Techniques would be a better term. A measurement is the action of measuring.

Page 3 line 49: measurement range sensitive enough Incorrect English: the measurement range is wide enough or the method is sensitive enough.

Page 5 - line 1: IpH should read pH

Page 7 - line 4: the word back-wash is missing between "treatment train, next"

Page 7 - line 6: in figure 1 and text describing this figure, it is stated that pressure drop is measured. In this sentence (page 7, line 6) the authors suggested it is not yet measured, but should be measured. Check for consistency and correctness.

Page 8: UV254 Hach 2018 it is strange to refer to the website of one manufacturer for a method which has been on the market for 20+ years and which is sold by a range of manufacturers. Please find a better reference for this. Example: https://iwaponline.com/ebooks/book/435/Compendium-of-Sensors-and-Monitors-and-Their-Use

this report contains an exhaustive overview of parameters for which online sensors were available at the time of publication.

Page 8: Surrogate parameters, UV254: Incorrect statement: measurement at all wavelengths is not a surrogate for UV254. If an instrument can measure across this range, it can also provide UV254, but the full spectrum is not a surrogate for UV254.

Page 8, bromate, Thermofisher 2018 Please explain this reference: referring to one manufacturer as proof that no online method exists is unconvincing. Please only indicate no (as above) which shows the authors have not found a method, or refer to an impartial review.

Page 8: ozone, Hach as comment for UV254.

Page 9: Phosphate, Hach as comment for UV254 and Ozone

Page 9: nitrogen please specific more precisely what parameter is meant here. Total nitrogen, Kjeldahl N, NO3?

Page 10 lines 10 -11: incorrect cause-effect relationship: the fact that the WHO published parameters does not mean sensors are available (as is suggested here). There are many parameters for which this is not the case. A more correct statement would simply be that for a number of WHO parameters sensors are available and then give some examples.

Page 12: i::scan Why mention specific product where it is only the parameter that is relevant. This product was not discussed before, nor are specific products described for turbidity and pH. Suggest to remove reference to specific product.

Page 13 line 6 and line 7: TM should be in superscript

Page 13 line 7: s::scan should be s::can or scan Messtechnik GmbH

―――――――――――――――――――

---

## Referee Comment (RC2) · Wim Audenaert (Referee) · 1 Jul 2019

GENERAL COMMENTS This manuscript can be descirbed as a highly applied research paper, bringing together formerly published concepts and applying them to two treatment units of specific drinking water treatment plant. Taking into account the importance of the topic, and the applied character of the journal, I am of the opinion that its publication will be valuable for many practitioners.

I agree with Referee 1 that a weakness of the paper is that real plant application and related improvements were not proven. However, coming back to the drinking water practitioners, the value lies in demonstrating the 7 step framework in a very easy way.

Most likely, not many drinking water plants have made such structural exercises, and this paper can lower the barrier of doing so. Hence, the objective of publishing this paper is not necessarily to present novel knowledge, but to show how to set up apply a monitoring framework in practice. It is important that the authors therefore reframe the paper as such, that it does not promise to provide the reader with a methodology that was proven to optimise a plant. It has the potential for that. The focus should be on illustration of practical use of such a framework, and the offering of a methodology to structurally question's one's train monitoring and control strategy.

SPECIFIC COMMENTS -There are very recent efforts going on with regard to on-line bromate sensor development, based on fluorescence measurement. This might be mentioned. The company Metawater is working on this (https://www.metawater.co.jp/eng/product/rd/sensor_technology/bromic_acid.html). -Fluorescence as a means of characterising NOM properties has not been mentioned. However, one-wavelength sensors are now being introduced on the market. Their benefit compared to UV-VIS might be their sensitivity at low DOM levels

TECHNICAL CORRECTIONS -Some references are missing in the reference list. Examples are Rieger et al., 2004; van der Helm et al., 2009. Please check for completeness. Potentially others are missing. -This paper is probably part of a PhD thesis. Remove any references to that, such as p4, line 27 ('Chapter') -typo at p4, line 16: 'imbedded' should be 'embedded' -p5, line 11: title should be Treatment step objectives, instead of treatment plant objectives -typo at p7, line 22: 'in the first columns' -typo at p7, line31: 'evaluation of available on-line sensors and their ...' p10, line24: 'cheap' –>describe more scientifically

---

## Author Comment (AC1) · 30 Nov 2019

The authors would like to thank the Referee for its thorough review and comments provided. We have reviewed the comments and below our remarks and adjustments made in the paper are indicated.

The paper "Design methodology to determine water quality monitoring strategy of surface water treatment plants" describes the application of a well-considered generic design strategy to analyse the possibilities and impossibilities of using available water quality sensors at one particular water treatment plant. The design strategy has been derived from a similar strategy developed for a control design methodology by

[Figure]

the (co)authors van Schagen en Rietveld (van Schagen et al in Water Science and Technology, 10, 121-127, 2010).

After a discussion of the aspects to be considered in each of the steps of assessment proposed, one case study is discussed in which an existing monitoring strategy was analysed and optimised.

Unfortunately, it is in the description of this case study and the subsequent conclusions, that the paper fails to address a number of essential elements that are necessary to judge the effectiveness of the design methodology.

We value the feedback received and acknowledge the fact that the paper the real plant application and related improvements were not proven. As a result we have chosen to modify the objective, discussion and conclusions of the paper to focus on a practical application for drinking water practitioners and the support the 7 step framework can give as a first step to develop an on-line water quality monitoring strategy in line with the suggestion of Referee 2.

Whereas sensor technology, characteristics of available types of instruments as well as treatment process characteristics and operational handles are described in substantial detail, no attention is giving to the following aspects: 1) although making selection a combination of sensors, and suggesting (page 13, lines 37-38) that this new combination was actually installed, the only comment the authors make about the results achieved is that more information is now available (page13 line 38) and that the balance between disinfection and by-product formation might be improved (page 13 line 49). The former is not an achievement of the presented methodology (any additional sensor installed would have resulted in more information becoming available) and the latter also follows from previous works by the authors, and but this is not confirmed in this study. Therefore, no improvement in process control or effluent water quality (the goals of this study, page 2, lines 31 - 37) are demonstrated.

The paper has been modified to not claim improvements in process control or water

quality but to demonstrate what a potential water quality monitoring strategy can be based on the gained understanding of the processes taking place.

2) As the authors present a method to determine (and / or optimise) a water quality monitoring strategy, it would be logical that the paper presents the results of the selected strategy and demonstrates that the outcome of the methodology actually produces an optimal monitoring setup. The authors do describe the selection process by describing both sensor options and processes (but not presenting any new information) but do not present any cost benefit analysis of the selected strategy, nor do they give any proof of how it improves the pre-existing monitoring setup at the location analysed. Therefore, the effectiveness of the design methodology cannot verified based on the results presented.

This was not the scope of the current paper. Although the authors agree that above would be a good next step to perform research on.

3) An implementation section, describing the experience with the actual deployment and operation of the sensors, this validation the methodology is missing. For an example, this reviewer would like to point the authors to the van Schagen paper (2010) from which the method described here was described. This paper has an implementation section where the approach is validated As result of the above the authors' conclusion (also found in the abstract) that the water quality programme of the Weesperkarspel plant was optimised is not sufficiently supported. The same is true for the conclusion on the use of soft-sensors (also mentioned in the abstract) - the authors state that the use of soft sensors helps to gain online information on parameters for which no online sensors are available. Although this may be correct, no new evidence to support this is presented in this paper. For this to be presented as one of the two major conclusions of the paper, it would be reasonable to expect the authors performed and analysis on the effectiveness of soft sensors in this type of application, and / or would have performed a cost benefit analysis (including CAPEX as well as OPEX costs) of using soft sensors vs. direct measurements.

This paper was focused on the first step to understand what could be options around on-line monitoring. The objective, discussion and conclusions have been modified to show how to set up a monitoring framework in practice and not promising to provide the reader with a methodology that was proven to optimise a plant.

Specific comments Page 1 - line 26: The authors appear to extrapolate the situation in the Netherlands to a generic statement (regarding sampling frequency). This is, however, incorrect. The sampling frequency depends on the size of the utility and the population served. It also depends on the parameter analysed. This statement currently is too generic. Either it should be restricted to the situation in the Netherlands (where all water companies / treatment plants are so large daily sampling is required or the water companies themselves take daily samples even if not required), or this statement should be rephrased taking into account daily practice also at smaller utilities in other countries.

Amended to primarily focus on the Netherlands.

Page 2 - line 3: time between sampling and analysis takes at least one day. The authors ignore the intermediate option of rapid tests that can be performed on-site. Utilities use such rapid tests for on-site measurements, e.g. to calibrate online sensors but also for data collection. Various parameters can even only be measured in this way (e.g. DO, O3, ...). It might be worth adding this type of analysis to the discussion, especially in view of the comment in the next sentence - such rapid tests can be used to verify performance.

The sentences have been modified to include the rapid tests and clarify above "Besides on-line measurements, laboratory measurements are taken at a regular interval to check the on-line measurements and that the produced drinking water meets the quality standards set by national and international guidelines. However, besides the rapid tests performed at Site, the time between sampling and laboratory results takes at least one day. This delay in results and interval between measurements makes it

difficult to only use the laboratory measurements for real-time control of a treatment plant".

Page 2 - lines 5 - 6: In addition, it should not be underestimated that erroneous control and measurement devices can also cause disturbances (van Schagen et al., 2010). It is not clear what the authors are trying to say with this sentence. Is the goal of this sentence to indicate that human operators can make mistakes and online sensors will detect this? Then remove the reference to erroneous measurement devices. If the goal is to stress the importance of correct use (installation, maintenance, ...) of online sensors, then this should be discussed separately. Currently, this sentence suggests online sensors are not trustworthy, whereas the next paragraph stresses the usefulness of such devices. These are now contradicting each other.

The sentence has been removed. The purpose of the sentence was to address the importance of correct use.

Page 3 - line 39: please reconsider this statement; fast degradation does not necessarily mean a need for high sensitivity and high accuracy. Sensitivity and accuracy will depend on where the measurements are taken and what the purpose of the measurement is. Examples: Does one want to verify the concentration dosed (at injection point) then the decay is less important and concentration will still be high. Downstream concentrations will be lower. but if initial concentration were high (e.g. with other chemical than ozone) still a not-so-sensitive sensor would still do the job. It seems that a fast measurement is most important for rapidly decaying ozone.

The statement has been changed to reflect the requirement of a fast measurement as is correctly pointed out above.

Page 4 - lines 26 - 28 Here it is stated the chapter focuses on mixed influent and mixed effluent for the processes ozonation and BAC filtration. However, in the preceding sentence the authors described the treatment plant consists of two parallel lanes. If this is correct, then the methodology appears to be incorrectly applied as it focused on the

non-existent situation of mixed ozonation effluent and mixed BAC influent. Could the authors clarify this and if this is indeed the case (2 treatment lanes and no combination of the water between ozonation and BAC filtration), please correct the text. This can be solved by rephrasing lines 26-28 and removing the reference to the mixed streams.

This has been amended into: "At Weepserkarspel, the production of drinking water is roughly divided into two parallel lanes (north and south lane), each consisting of several individual reactors/filters per treatment step. In each lane the water is mixed after each treatment step. The control actions can be modified at individual level, however, for the purpose of this paper it has been chosen to focus on the mixed influent and effluent of one lane only and not on the individual reactor/filter level.

Page 5 - lines 21 - 22: The authors state that the adsorption capacity (of the carbon filters) decreases with increasing polarity (of the organic matter). It appears that the equilibrium between adsorption and desorption will shift and the affinity of more polar organic matter for the carbon filter surface will be reduced. However, the adsorption capacity of the carbon (the maximum amount of material it can adsorb) should not be affected, as the ozonation does not change the surface properties of the adsorbent.

The adsorption capacity has been modified into adsorption affinity. Furthermore it has been added that due to the pre-oxidation step, the NOM is not only removed through adsorption but in additional also through biodegradation. In addition please see the response to below question.

Page 5 - lines 22 - 23: As a result NOM is removed through ... adsorption. The current wording states that because NOM is oxidised and becomes more polar it is removed through biodegradation and adsorption. Is this true, i.e. would NOM be removed only through biodegradation or adsorption if no ozonation takes place?

If no ozonation or peroxidation would have taken place, the main removal mechanism of NOM in Activated carbon would be through adsorption also referred to as Granular Activated Carbon (GAC) filtration. By enhancing the biodegradability due to oxidation

of the NOM into smaller organic molecules, biodegradation is strongly promoted. It is not so much the polarity, but more the biodegradable organic matter that is formed after oxidation, which enhances biodegrdation.

Page 5 - lines 33 - 34: Therefore, ... control action. This appears to be a missed opportunity. The authors indicate the flow through the plant is variable (because determined by the demand for drinking water). This means the flow through the treatment lanes is variable. Keeping flow constant could be a very useful control measure for the performance of individual treatment steps. Changes the division of the water over the different production lines, (although not used currently) could thus be used to manage the performance of the treatment processes, and optimise it. E.g. though keeping one lanes at ideal operational conditions, which might result in better net quality of the mixed effluent. It would have been worth investigating this possibility. Eliminating this option at the beginning of the evaluation was a missed opportunity. Did the authors consider this type of evaluation?

Due to the presence of clean water storage reservoirs a certain buffer capacity is guaranteed to overcome strong fluctuations by the changes in demand for drinking water. The text has been modified to clarify this. As a result of this buffering capacity, the treatment lanes are typically operated at a constant flow, hence no further optimization is required at this stage. The following sentence has been added: "Due to the buffering capacity of the clean water storage reservoirs, the treatment plant is already operated at a constant optimized flow, therefore, in this case, production flow was not considered as a control action." Have the authors considered this possibility? Is there information (other than the fact that the plant operators do not use this control option) that it would not be been relevant? Please see above.

Page 5 - lines 45-46. Because pH is important for the functioning of the BAC filters, can one be sure that the influent water (the effluent of the pellet softeners) is always at the correct / optimal pH? If this is the case, it is correct to disregard this parameter for control of the BAC. If the pH can be variable (e.g. because there is no real-time control

of the CO2 dosing), then pH control should be seen as a control action for the BAC, even though the actual dosing equipment might be part of the pellet softener. Please add a statement that explains why it is not necessary to adjust pH before BAC, e.g. because it is always the same or because it is actively controlled in the softening.

The pH is actively controlled as part of the softening step. The following sentence has been modified to reflect this: "Dosing of carbon dioxide is actively controlled based on the measured pH This control action is thus morewhich is related to the functioning of the pellet softeners and therefore not included in the overview provided in Figure 1."

Page 6 - lines 14 - 15: Monitoring of TOC/DOC/SUVA will only be necessary if the composition of the NOM changes. The authors describe the water is seepage water, which is groundwater. Please explain why this variable is relevant for the operation of this treatment plant: is there a variable composition in NOM that would necessitate monitoring in this example.

The feed water is mainly humics' rich seepage water which is sometimes mixed with Amsterdam-Rhine canal water. When looking at the ozone influent the DOC fluctuates between 4.5 – 8 mg-C/L and the SUVA between 1-3 ((1/m)/mg-C/L). The mixing with Amsterdam-Rhine canal water has been added to the case study description.

Page 6 - line 28: Bromate is possibly or probable carcinogenic. Suggest to refer to only one reference that reflects the latest insights into this topic. Referring to both possibly and probable is confusing.

Amended.

Page 6 - line 38: ATP or flowcytometry. Please explain this statement. There was no reference before to these methods and as to why they would be required. Therefore, the statement on the absence of a need for these methods is confusing and seems irrelevant.

Amended to show that no on-line measurement of bacterial cells is required which

was shown by Ross et al. (2019)., instead of creating confusing by translating the measurement of bacterial cells by the mentioning of possible methods that can be used.

Page 8 - bromide: bromide sensors (ion selective electrodes) are available from various manufacturers. Please correct.

Amended

Page 8, Ozone, soft sensor available. Yes, developed based on UV measurement The measurement of ozone using UV spectroscopy is a direct measurement; O3 has a distinct spectrum which is directly measured. In the case of a soft sensor a number of measurements are taken together to estimate a parameter which can not directly be measured. For O3 by UV spectroscopy this is explicitly not the case, as the O3 spectrum itself is directly measured.

Amended.

Page 12 - lines 27-28: When evaluating... it is not required to monitor ... due to robustness of the BAC filtration step. As the fact that it is not necessary to monitor the AOC was already known from previous work by the authors, why did they ignore this knowledge in the preceding discussion. This type of prior knowledge should flow into the design methodology as early as possible, as it prevents a waste of time (e.g. in this case the discussion on AOC could have been skipped).

The way the methodology is set-up is to first focus on the individual treatment steps and subsequently performs the final check if any measurements can be removed based on the integral assessment of the treatment plant. Therefore in this specific case it turned out that it was not required to monitor the AOC due to the robustness of the BAC filters in Weesperkarspel. For any other plant this should be verified and might not be the case.

Technical corrections Page 3 line 43: in "A wide range of measurement" Methods or

Methodologies or Techniques would be a better term. A measurement is the action of measuring.

Amended into 'methodologies'

Page 3 line 49: measurement range sensitive enough Incorrect English: the measurement range is wide enough or the method is sensitive enough.

Amended into 'method sensitive enough'

Page 5 - line 1: IpH should read pH - Amended Page 7 - line 4: the word back-wash is missing between "treatment train, next" - Amended Page 7 - line 6: in figure 1 and text describing this figure, it is stated that pressure drop is measured. In this sentence (page 7, line 6) the authors suggested it is not yet measured, but should be measured. Check for consistency and correctness.

In the Results section it is described that based on the theory the pressure drop should be measured. In the used example (Figure 1) this happens to be the case, hence the described practice matches the theory.

Page 8: UV254 Hach 2018 it is strange to refer to the website of one manufacturer for a method which has been on the market for 20+ years and which is sold by a range of manufacturers. Please find a better reference for this. Example:https://iwaponline.com/ebooks/book/435/Compendium-of-Sensors-and-Monitorsand-Their-Use this report contains an exhaustive overview of parameters for which online sensors were available at the time of publication.

Suggested reference included and replaced for the manufacturer references.

Page 8: Surrogate parameters, UV254: Incorrect statement: measurement at all wavelengths is not a surrogate for UV254. If an instrument can measure across this range, it can also provide UV254, but the full spectrum is not a surrogate for UV254.

Amended

Page 8, bromate, Thermofisher 2018 Please explain this reference: referring to one manufacturer as proof that no online method exists is unconvincing. Please only indicate no (as above) which shows the authors have not found a method, or refer to an impartial review.

Amended

Page 8: ozone, Hach as comment for UV254.

Amended, suggested reference as above has been included

Page 9: Phosphate, Hach as comment for UV254 and Ozone - Amended Page 9: nitrogen please specific more precisely what parameter is meant here. Total nitrogen, Kjeldahl N, NO3?

Kjeldahl-N, amended in text

Page 10 lines 10 -11: incorrect cause-effect relationship: the fact that the WHO published parameters does not mean sensors are available (as is suggested here). There are many parameters for which this is not the case. A more correct statement would simply be that for a number of WHO parameters sensors are available and then give some examples.

Reference to WHO has been removed. Sentence has been modified to 'There are sufficient on-line sensors available to measure the pH, temperature and DO.'

Page 12: i::scan Why mention specific product where it is only the parameter that is relevant. This product was not discussed before, nor are specific products described for turbidity and pH. Suggest to remove reference to specific product.

Reference in Figure 3 has been removed, in the text it was kept in to indicate that Weesperkarspel had chosen to specifically focus on UV254 instead of the full spectrum.

Page 13 line 6 and line 7: TM should be in superscript - Amended Page 13 line 7:

[Figure]

s::scan should be s::can or scan Messtechnik GmbH - Amended into 's::can'

Please also note the supplement to this comment:
https://www.drink-water-eng-sci-discuss.net/dwes-2019-8/dwes-2019-8-AC1-supplement.pdf

―――――――――――――

---

## Author Comment (AC2) · 30 Nov 2019

The authors would like to thank the Referee for its thorough review and comments provided. We have reviewed the comments and below our remarks and adjustments made in the paper are indicated.

GENERAL COMMENTS This manuscript can be descirbed as a highly applied research paper, bringing together formerly published concepts and applying them to two treatment units of specific drinking water treatment plant. Taking into account the importance of the topic, and the applied character of the journal, I am of the opinion that its publication will be valuable for many practitioners.

[Figure]

I agree with Referee 1 that a weakness of the paper is that real plant application and related improvements were not proven. However, coming back to the drinking water practitioners, the value lies in demonstrating the 7 step framework in a very easy way. Most likely, not many drinking water plants have made such structural exercises, and this paper can lower the barrier of doing so. Hence, the objective of publishing this paper is not necessarily to present novel knowledge, but to show how to set up apply a monitoring framework in practice. It is important that the authors therefore reframe the paper as such, that it does not promise to provide the reader with a methodology that was proven to optimise a plant. It has the potential for that. The focus should be on illustration of practical use of such a framework, and the offering of a methodology to structurally question's one's train monitoring and control strategy.

We have modified the objective, discussion chapter and conclusions of the paper in line with above suggestions. The objective has been changed to: "Therefore, in this paper a design methodology is described which helps to develop a water quality monitoring scheme. This will be explained by means of a case study for the WTP Weesperkarspel in the Netherlands."

The discussions have been changed to address advances in on-line water quality monitoring, reliability of the data and on-line water quality monitoring strategy instead of direct control based on water quality.

The first paragraph of the conclusions have been changed to: "The main objective of this paper was to develop a design methodology supporting the development of a water quality monitoring strategy. A seven step approach was defined, and each step was demonstrated for the treatment processes ozone and BAC filtration. It was shown how the previous on-line water quality monitoring program of the treatment plant Weesperkarspel was adjusted based on a better understanding of the processes taking place.

SPECIFIC COMMENTS There are very recent efforts going on with regard

to on-line bromate sensor development, based on fluorescence measure-
ment. This might be mentioned. The company Metawater is working on this
(https://www.metawater.co.jp/eng/product/rd/sensor_technology/bromic_acid.html).
Fluorescence as a means of characterising NOM properties has not been mentioned.
However, one-wavelength sensors are now being introduced on the market. Their
benefit compared to UV-VIS might be their sensitivity at low DOM levels

We have included a general message on ongoing developments and chosen to include
only references to published work. A reference to fluorescence has been included in
the paper in the section required water quality parameters addressing the characteri-
zation of NOM.

TECHNICAL CORRECTIONS Some references are missing in the reference list. Ex-
amples: are Rieger et al., 2004; van der Helm et al., 2009. Please check for complete-
ness. Potentially others are missing. -This paper is probably part of a PhD thesis.

Amended, full paper has been checked and missing references added.

Remove any references to that, such as p4, line 27 ('Chapter') Amended to 'paper'
typo at p4, line 16: 'imbedded' should be 'embedded' - Amended p5, line 11: title
should be Treatment step objectives, instead of treatment plant objectives: - Amended
typo at p7, line 22: 'in the first columns' - Amended typo at p7, line31: 'evaluation
of available on-line sensors and their ...' - Amended p10, line24: 'cheap' –>describe
more scientifically - Amended to low cost

Please also note the supplement to this comment:
https://www.drink-water-eng-sci-discuss.net/dwes-2019-8/dwes-2019-8-AC2-
supplement.pdf
* * *